# Investigating the Impact of Carbon Fiber as a Wheelchair Frame Material on Its Ability to Dissipate Kinetic Energy and Reduce Vibrations

**DOI:** 10.3390/ma17030641

**Published:** 2024-01-29

**Authors:** Bartosz Wieczorek, Łukasz Warguła, Jarosław Adamiec, Tomasz Sowa, Michał Padjasek, Łukasz Padjasek, Maciej Sydor

**Affiliations:** 1Faculty of Mechanical Engineering, Institute of Machine Design, Poznan University of Technology, Piotrowo 3, 60-965 Poznań, Poland; bartosz.wieczorek@put.poznan.pl (B.W.); jaroslaw.adamiec@put.poznan.pl (J.A.); 2Scientific and Research Centre for Fire Protection, National Research Institute, Nadwiślańska 213, 05-420 Józefów, Poland; tsowa@cnbop.pl; 3Cosmotech LLC, Szyby Rycerskie 22k St., 41-909 Bytom, Poland; mp@cosmotech-3d.com (M.P.); lp@cosmotech-3d.com (Ł.P.); 4Department of Woodworking and Fundamentals of Machine Design, Faculty of Forestry and Wood Technology, Poznań University of Life Sciences, 60-637 Poznań, Poland; maciej.sydor@up.poznan.pl

**Keywords:** vibration damping, vibrations, damping decrement, vibration transmission, impact on the human body

## Abstract

Using a wheelchair over uneven terrain generates vibrations of the human body. These vibrations result from mechanical energy impulses transferred from the ground through the wheelchair components to the user’s body, which may negatively affect the quality of the wheelchair use and the user’s health. This energy can be dissipated through the structure of the wheelchair frame, such as polymer and carbon fiber composites. This article aims to compare a wheelchair with an aluminum alloy frame and a carbon fiber frame in terms of reducing kinematic excitation acting on the user’s body. Three wheelchairs were used in the study, one with an aluminum alloy frame (reference) and two innovative ones with composite frames. The user was sitting in the tested wheelchairs and had an accelerometer attached to their forehead. The vibrations were generated by applying impulses to the rear wheels of the wheelchair. The obtained results were analyzed and compared, especially regarding differences in the damping decrement. The research shows that using modern materials in the wheelchair frame has a beneficial effect on vibration damping. Although the frame structure and material did not significantly impact the reduction in the acceleration vector, the material and geometry had a beneficial effect on the short dissipation time of the mechanical energy generated by the kinematic excitation. Research has shown that modern construction materials, especially carbon fiber-reinforced composites, may be an alternative to traditional wheelchair suspension modules, effectively damping vibrations.

## 1. Introduction

A person using a wheelchair forms an anthropotechnical system that can be modeled with many related biomechanical and mechanical parameters and causal effects [1,2,3]. One such parameter relation is the movement of the human body combined with the movement of a wheelchair [4,5]. This connection results from the cyclical propelling of the wheelchair using the upper limbs and the transfer of mechanical energy to the human body through the body support system of the moving wheelchair [6]. Negotiating uneven terrain can introduce additional vertical accelerations into the wheelchair–user system, particularly when the wheelchair abruptly transitions between different height levels, such as rolling off a curb [7,8,9]. These abrupt changes can generate unfavorable vibrations that significantly impact the user’s comfort and stability. The source of these vibrations are stochastic impulses of mechanical energy transmitted from the ground to the human body via the wheels, frame, and seat of the wheelchair. These impulses cause short-term and sudden accelerations affecting the body and internal organs [10,11].

Human body vibrations are an unfavorable phenomenon that deteriorates the quality of wheelchair use and poses a risk to its user’s health [12,13,14,15]. Therefore, the need to achieve the ability to dampen wheelchair vibrations that occur during driving is justified. The current state of technology allows for solving this problem with additional wheelchair equipment. Known solutions include shock-absorbing cushions [16,17,18] and shock-absorbing wheelchair suspension [19,20,21]. A new trend is using innovative geometric features of the wheelchair frame structure [22] or new materials that absorb vibration energy [18]. An example of a new material used in the construction of wheelchair frames is polymer and carbon fiber composites. So far, this material has been used due to a significant reduction in the weight of the wheelchair [23,24,25] while maintaining high stiffness, as well as due to its modern visual design. 

Analyzing the available research works, there is a noticeable lack of studies analyzing the impact of carbon fiber composites in a wheelchair frame structure on the reduction in vibrations transmitted to the human body. Therefore, this study seeks to compare the shock-absorbing capabilities of a wheelchair with an aluminum alloy frame and one with a carbon fiber frame, focusing on reducing the magnitude and duration of accelerations experienced by the wheelchair user’s body. 

Achieving this goal required the preparation of a consistent vibration generator and the analysis of the acceleration values acting on the head of the wheelchair user and the time needed to reduce these accelerations to the level of 1 g (approximately 9.81 m/s^2^). Based on the tests performed, it is possible to determine the ability of new-generation wheelchairs to dampen mechanical energy caused by kinematic excitation coming from the ground. 

## 2. Materials and Methods

### 2.1. Tested Wheelchairs

We used three ultra-lightweight rigid manual self-propelled wheelchairs, one with an aluminum frame and two with a carbon fiber frame. Table 1 summarizes the tested wheelchairs: WA denotes a wheelchair with an aluminum frame, WCBK signifies a model with a carbon fiber frame and a standard seat, while WCK represents a configuration featuring a carbon fiber frame and a bucket seat.

The WA, WCK, and WCBK wheelchair variants built for this study had the same running gear (drive wheels and caster wheels) and identical anti-decubitus foam cushions. The differences between the commonly used WA wheelchair with an aluminum frame and new carbon fiber (WCK and WCBK) structures resulted from the frame’s material, cross-sections, and seat type (Figure 1).

During the tests, the same person was always sitting in a wheelchair: a man with a height of 176 cm and a body weight of 64 kg (the person sat motionless in an upright position). An external generator generated the vibrations. The wheelchairs were stabilized during the tests and did not move.

### 2.2. Research Equipment

The research was carried out using the measurement system shown in Figure 2, consisting of the tested wheelchair (1), whose front self-adjusting wheels were supported on a stationary surface (2), while the rear wheels were based on a cylinder (3) with a diameter equal to the diameter of the wheelchair’s drive wheel (600 mm). The cylinder (3) was coupled to the wheel by an electric motor generating a constant rotational speed of 20 rpm, which translated into a simulated linear speed of the wheelchair of approximately 2.28 km/h. A transverse irregularity (4) with a height of 15 mm was placed on the cylinder (3), which was a generator of vibrations transmitted to the tested human wheelchair anthropotechnical system. The amplitude of each generated vibration pulse was 15 mm, and the pulse acceleration was 5.36 g.

The element recording vibrations (in the form of *R* acceleration) was an accelerometer (6) mounted on the person’s head, attached with elastic bands, pressed to the Vertex capitis head, and stabilized in this position. The accelerometer used consisted of a KIONIX kx023 inertial sensor (with a resolution of 0.009576801 m/s^2^ and a measurement range from 0 to 78.4532 m/s^2^), a data archiving system, and a power supply. The mass of the vibration recorder did not exceed 100 g, so it was insignificant compared to the overall weight of the anthropotechnical systems evaluated. The layer adapting the flat surface of the bottom of the accelerometer to the curved surface of the head of a person sitting in the tested wheelchair was made of rubber with a hardness of 40–50 according to the Shore A scale. This method of mounting the vibration recorder is consistent with the guidelines contained in the PN-EN ISO standard 5349-1:2004 [26] for the supporting frame to which the drive unit and supports holding the entire device on the ground were also attached. The position of the accelerometer’s X, Y, and Z measurement axes relative to the frame of the tested wheelchairs is shown in Figure 2.

### 2.3. Analytical Model

The analytical model of the studied system, with an indication of the analyzed element in terms of energy dissipation properties, is depicted in Figure 3. This model divides the studied system into components: front wheel, rear wheel, wheelchair frame, human body, and head. Such a segmented division of the system and the connections between segments are consistent with scientific works dedicated to modeling the vibrations of a wheelchair. The last segment, the head, provided the measured signal during the study, which was the subject of the analysis of the influence of the frame material on the dissipation of acceleration transferred to the human body.

Based on the above schematic, analytical equations describing the displacements of individual elements of the system (1) can be formulated as follows:(1)$$\left[\begin{array}{cc} \begin{array}{ccc} m_{1} & 0 & 0\\ 0 & m_{2} & 0\\ 0 & 0 & m_{3} \end{array} & \begin{array}{ccc} 0 & 0 & 0\\ 0 & 0 & 0\\ 0 & 0 & 0 \end{array}\\ \begin{array}{ccc} 0 & 0 & 0\\ 0 & 0 & 0\\ 0 & 0 & 0 \end{array} & \begin{array}{ccc} \ddot{I} & 0 & 0\\ 0 & m_{4} & 0\\ 0 & 0 & m_{5} \end{array} \end{array}\right]\left[\begin{array}{c} \begin{array}{c} \ddot{x_{1}}\\ \ddot{x_{2}}\\ \ddot{x_{3}} \end{array}\\ \begin{array}{c} \ddot{\Phi }\\ \ddot{x_{4}}\\ \ddot{x_{5}} \end{array} \end{array}\right]\;+\left[\begin{array}{cc} \begin{array}{ccc} C_{1} & -C_{1} & 0\\ C_{1} & C_{1}+C_{2} & -C_{2}\\ 0 & -C_{2} & C_{2}+C_{3}+C_{4} \end{array} & \begin{array}{ccc} 0 & 0 & 0\\ C_{1}\left(-L_{2}+L_{3}\right) & 0 & 0\\ C_{2}\left(L_{2}-L_{3}\right)-C_{3}L_{1}+C_{4}L_{2} & -C_{3} & -C_{4} \end{array}\\ \begin{array}{ccc} 0 & -C_{2}L_{3} & C_{2}L_{3}+C_{3}L_{1}+C_{4}L_{2}\\ 0 & 0 & -C_{4}\\ 0 & 0 & -C_{5} \end{array} & \begin{array}{ccc} C_{2}L_{2}-C_{2}L_{3}^{2}-C_{3}L_{1}^{2}+C_{4}L_{2}^{2} & {-C}_{3}L_{1} & -C_{4}L_{2}\\ C_{4}L_{1} & C_{4}+C_{6} & 0\\ -C_{5}L_{2} & 0 & C_{5}+C_{7} \end{array} \end{array}\right]\left[\begin{array}{c} \begin{array}{c} \dot{x_{1}}\\ \dot{x_{2}}\\ \dot{x_{3}} \end{array}\\ \begin{array}{c} \dot{\Phi }\\ \dot{x_{4}}\\ \dot{x_{5}} \end{array} \end{array}\right]\;+\left[\begin{array}{cc} \begin{array}{ccc} K_{1} & -K_{1} & 0\\ K_{1} & K_{1}+K_{2} & -K_{2}\\ 0 & -K_{2} & K_{2}+K_{3}+K_{4} \end{array} & \begin{array}{ccc} 0 & 0 & 0\\ K_{1}\left(-L_{2}+L_{3}\right) & 0 & 0\\ K_{2}\left(L_{2}-L_{3}\right)-K_{3}L_{1}+K_{4}L_{2} & -K_{3} & -K_{4} \end{array}\\ \begin{array}{ccc} 0 & -K_{2}L_{3} & K_{2}L_{3}+K_{3}L_{1}+K_{4}L_{2}\\ 0 & 0 & -K_{4}\\ 0 & 0 & -K_{5} \end{array} & \begin{array}{ccc} K_{2}L_{2}-K_{2}L_{3}^{2}-K_{3}L_{1}^{2}+K_{4}L_{2}^{2} & {-K}_{3}L_{1} & -K_{4}L_{2}\\ K_{4}L_{1} & K_{4}+K_{6} & 0\\ -K_{5}L_{2} & 0 & K_{5}+K_{7} \end{array} \end{array}\right]\left[\begin{array}{c} \begin{array}{c} x_{1}\\ x_{2}\\ x_{3} \end{array}\\ \begin{array}{c} \Phi \\ x_{4}\\ x_{5} \end{array} \end{array}\right]\;=\left[\begin{array}{c} \begin{array}{c} 0\\ 0\\ F \end{array}\\ \begin{array}{c} F\cdot L_{3}\\ 0\\ 0 \end{array} \end{array}\right]$$

### 2.4. Signal Processing Research Methodology

The research procedure assumed mounting the tested wheelchair in a system generating kinematic excitations (Figure 2) [27]. This system generated a sudden vertical displacement of the rear wheels of the wheelchair to a height of 15 mm in 3 s intervals. This method of generating kinematic excitations is referenced in the works of other researchers using a round rod with a diameter of 3/8” (9.53 mm) to generate vibrations of a similar amplitude and frequency [28]. This interaction resulted in the excitation of vibrations that propagated from the wheelchair’s drive wheels to the frame and seat, ultimately reaching the study participant’s body. The numerical interpretation of the generated vibrations was expressed as an *R* function of the acceleration time of the accelerometer mounted on the head of a person sitting in a wheelchair. The procedure for processing the measurement signal is shown in Figure 4. According to this algorithm, the value of the *R* resultant acceleration measured with an accelerometer as a function of time was used to analyze vibrations. 

Five measurement samples were isolated from the measured *R* acceleration as a function of time course, discarding the extreme values (A1). Then, seven consecutive maximum values (A2) of the *R* resultant acceleration were searched for each isolated sample of cyclically repeating kinematic excitation. On this basis, a set of points representing the time *t*_n_ and the amplitude *A*_n_ corresponding to this time were defined. Based on this set, the variability of the *R* acceleration amplitude as an A6 function of time was first analyzed. This analysis determined the average *A* value of the amplitudes (2) from all isolated measurement samples. And this average was represented as a function of the normalized *t*_norm_ excitation duration (3). A graphical interpretation of these arithmetic operations below is shown in Figure 5.
(2)$$avg.\;A_{n}=\frac{\sum_{i=1}^{i=5}A_{n,i}}{5}$$
(3)$$t_{norm}^{n}=\frac{\sum_{i=1}^{i=5}\left(t_{n,i}-t_{n-1,i}\right)}{5}$$
where *n*—number of the cycle in the analyzed measurement sample of kinematic excitation, *i*—number of the separated measurement sample, *A*—amplitude value, *t*—time value for the performed amplitude observation.

In the next stage of the performed analysis, the value of the *δ*_n_ damping decrement (Figure 4(A5)) (4) [29] was calculated for the subsequent *A*_i_ amplitudes in relation to the first and maximum amplitude value (Figure 4(A1)) in the cycle selected for analysis from the recorded measurement test.
(4)$$\delta_{n}=\frac{A_{0}}{A_{n+1}}\;;n=0,\;\ldots ,\;6$$
where *δ_n_*—an *n*-th value of the damping decrement for the analyzed kinematic excitation cycle, *A*_1_—a value of the first amplitude of the analyzed kinematic excitation cycle, *A_n_*_+1_—second and subsequent values of the amplitude of the analyzed kinematic excitation cycle.

The determined damping decrement values for five measurement samples representing separate cycles of the measured kinematic excitation (Figure 4(A4)) were then averaged, and the limits of the confidence interval were calculated using the student’s t-distribution and the confidence level *p* = 0.05 (Figure 4(A7)). This average value of the damping decrement was used to analyze the amplitudes after reaching the point at which the damping decrement value stabilizes at an equal level. Stabilization of the damping decrement was considered fulfilled if the value of the nth δ_n_ damping decrement was in the range of <δ_6_ − 0.5%; δ_6_ + 0.5%> (Figure 5), and when at the same time the value of the measured *R* acceleration vector was approximately 1 g (approximately 9.81 m/s^2^), which is the value that naturally affects the wheelchair at rest in the force field of Earth’s gravity. Figure 6 illustrates the stabilization of the damping decrement over a specific range.

## 3. Results and Discussion

The results of the analysis of vibration amplitude values for the three tested wheelchairs are presented in Figure 7 and Table 2. Upon analyzing the acquired results, it was observed that the value of the first *A*_1_ amplitude of the *R* vector, occurring after the initiation of kinematic excitation, consistently ranged from 10.85 to 10.95 m/s^2^, regardless of the frame structure of the wheelchair, which constitutes a slight difference of only 0.1 m/s^2^. Additionally, a similar increase in the acceleration value acting on the head was observed in other scientific works using a similar method of generating kinematic excitation [30]. In his research, Philip S. Requejo generated acceleration acting on the user’s head, reaching a maximum of 12 m/s^2^ [28]. A slight difference in this value may result from differences in the design features of the excitation generator (steel cylinder on which the wheelchair drive wheel rolled) and design differences of the tested wheelchairs. The importance of the influence of the generator of kinematic excitations type is confirmed by research carried out by Philip S. Requejo, who, in his subsequent studies, generated kinematic excitations during a wheelchair descent from a curb [31]. In this case, the acceleration value increased significantly, reaching values from 1.69 to 1.33 G.

The analysis of the A_7_ amplitude of the *R* vector, i.e., the seventh amplitude after the kinematic excitation initiation, showed that regardless of the type of the tested wheelchair, the acceleration acting on the human head reaches a value similar to the acceleration corresponding to a wheelchair at rest located in the Earth’s gravitational field. When the A_7_ amplitude was reached, the value of the *R* vector ranged from 9.87 m/s^2^ to 9.90 m/s^2^, which is a difference of 0.03 m/s^2^. Additionally, this result is close to the acceleration value due to gravity, proving that the initiated kinematic excitation is damped after a time equal to seven times the *T* period. In the tested cases, this time was approximately 1.5 s for the average *T* = 0.250 s.

The analysis of vibration amplitudes for wheelchairs made of carbon fiber composite (WCBK and WCK) showed that its value in the first oscillations is almost equal despite the difference in the seat system design. The analysis of the acceleration oscillations showed that in the case of a wheelchair with an aluminum alloy frame (WA) for the time interval from 0.4 to 1.2 s, the amplitude value is higher than for carbon fiber wheelchairs (WCBK and WCK). The peak value of the difference in the value of the vibration *R* vector in these amplitudes is 0.75 m/s^2^, which occurs in the time interval from 0.6 to 0.8 s. This observation is confirmed by the work describing the impact of the construction material and construction features on the transmission and damping of vibrations [32,33,34,35].

The impact of long-term vibrations and sudden accelerations on the human body is unfavorable due to the risk of health deterioration [36,37]. Therefore, a beneficial operating property of a wheelchair is the shortest possible time needed to reduce the acceleration acting on the human body to a level close to 1 g. Consequently, further research analyzed the value of the damping *δ* decrement and observed when its value stabilized over time [38]. The constant value of *δ* as a function of time indicates that the analyzed value of the A_i_ acceleration amplitudes has stabilized (Figure 8). The adopted research methodology translated into a reduction in the acceleration acting on the head to approximately 1 g. The analysis of the damping decrement of the tested wheelchairs was performed based on five separate time courses of the *R* vector. Each separated signal began until the kinematic excitation acted on the tested system and lasted seven times the *T* period, allowing to observe the n number of deflections of the *A*_i_ amplitude, equal to *n* = 7.

The analysis results of the *δ* damping decrement indicate that in the case of the WA wheelchair, the initial value of the damping decrement is much smaller than in the case of the WCBK and WCK wheelchairs. The value of δ for the oscillation corresponding to n = 1 is in the ranges: from 1.028 to 1.070 for the WA wheelchair, from 1.059 to 1.090 for the WCBK wheelchair, and from 1.057 to 1.083 for the WCK wheelchair. By analyzing the function that describes the variability of the average damping decrement *δ*, we found that in the case of wheelchairs made of carbon fiber composite (WCBK and WCK), the curve showing the change in damping decrement flattens much earlier than in the case of the WA wheelchair. Therefore, another analysis was performed, this time checking the variability of the average value of the δ_avg_ damping decrement as a function of the number of the analyzed n oscillations (Figure 9).

In the *δ*_avg_ analysis performed following the research methodology, it was assumed that the value of the damping decrement of subsequent amplitudes (from A_1_ to A_n_) is constant if it falls within the range of δ(*n* = 6) ± 0.5%. This assumption means that the ratio of the first measured amplitude to the *n*-th consecutive amplitude is approximately equal (5).
(5)$$\frac{A_{0}}{A_{n}}\approx \frac{A_{0}}{A_{n+1}}\approx \frac{A_{0}}{A_{n+2}}\;↔\sigma_{n}\approx \sigma_{n+1}\approx \sigma_{n+2}$$

The average *δ*_avg_ damping decrement analysis confirmed previous observations stating that WCBK and WCK wheelchairs dampen the initiated kinematic excitation more quickly. This phenomenon results from using a new material for this type of construction, i.e., carbon fiber composite, which is confirmed in the literature [39]. For the WA wheelchair, stabilization of δ_avg_ occurred after the *t*_norm_ time of 1.268 s with a total duration of the kinematic excitation of 1.498 s. This constituted approximately 85% of the entire duration of the kinematic excitation. In the case of the WCBK and WCK wheelchairs, the stabilization of δ_avg_ occurred after the *t*_norm_ time, which was 1.117 s and 0.868 s, respectively. This constituted 74% of the total duration of the kinematic excitation for the WCBK wheelchair and 58% for the WCK. Based on the direct measurement of vibration acceleration, it is possible to indirectly determine, for instance, the mechanical energy transferred to the human body. Similar acceleration calculations allowing the determination of mechanical and energy parameters are carried out in many fields of science [40,41,42,43].

## 4. Conclusions

The conducted research has shown that the use of modern materials to construct a wheelchair frame requires an interdisciplinary approach to the design process, which also takes into account factors such as the ability to dampen vibrations. The use of materials or design solutions that effectively reduce vibrations translates directly into the comfort of using the wheelchair and reduces the risk to the health of its user. The kinematic excitation used in the study resulted in a linear acceleration acting on the human head ranging from 10.85 to 10.92 m/s^2^. Using the same method of generating kinematic excitation for all tested wheelchairs, similar values of the vibration amplitude measured on the wheelchair user’s head in the zero cycle (A_0_) were obtained. Therefore, it can be concluded that the frame structure and the material from which the tested wheelchairs were made do not reduce the value of the R acceleration vector. 

Based on the performed tests, it was also found that the material and geometry of the tested wheelchairs only influenced the time of dissipation of mechanical energy caused by kinematic excitation. During the WA wheelchair test, the initiated kinematic excitation decreases to the value of 9.81 m/s^2^, corresponding to the acceleration due to gravity after performing five analyzed oscillations. However, in the case of WCBK and WCK wheelchairs, the kinematic excitation was reduced to 9.81 m/s^2^ after just three analyzed oscillations. It should be noted that seven consecutive oscillations were used in the analyses, the first of which, marked with the number 0, was the reference value for calculating the damping decrement. The results confirm that modern construction materials can be an alternative to additional wheelchair suspension modules that dampen vibrations caused by kinematic excitations. 

Reducing the duration of dynamic overloads affecting the human body is a significant problem that translates directly into the health of the wheelchair user. The research showed that one of the possible solutions to this problem is using modern engineering materials, such as carbon fiber-reinforced composites. The performed damping decrement analysis showed an increase of 10% in the rate of dissipation of mechanical energy generating vibrations of the human body for the WCBK wheelchair compared to the structure of a wheelchair with an aluminum alloy frame (WA) and an increase of 32% in the rate of mechanical energy dissipation for WCK wheelchair compared to the structure of a wheelchair with an aluminum frame (WA). WCBK and WCK wheelchairs, despite better results compared to the WA wheelchair, show a significant difference in results. These wheelchairs had the same structure of the supporting frame and drive system elements but differed in the seat arrangement. Therefore, further research should assess how individual elements of the wheelchair structure made of carbon fiber affect the kinematic excitation operation time reduction. 

## Figures and Tables

**Figure 1 materials-17-00641-f001:**
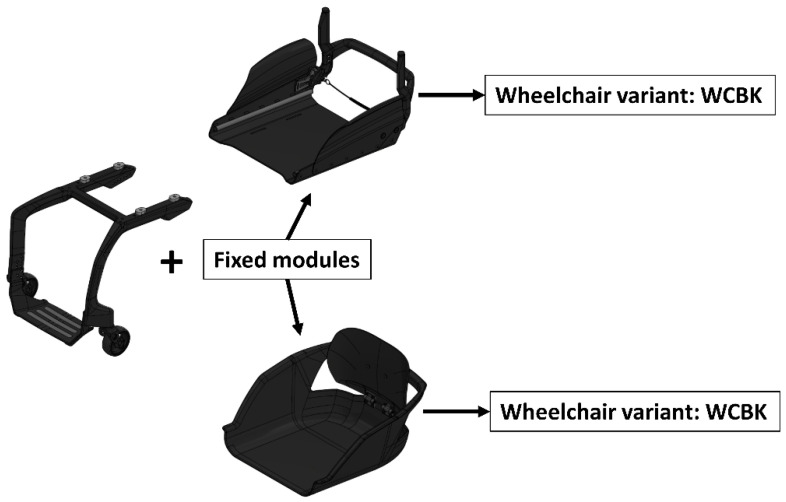
Configuration diagram of the Cosmotech Freeasy classic (WBCK) and Cosmotech Freeasy bucked (WCK) carbon fiber wheelchair modules.

**Figure 2 materials-17-00641-f002:**
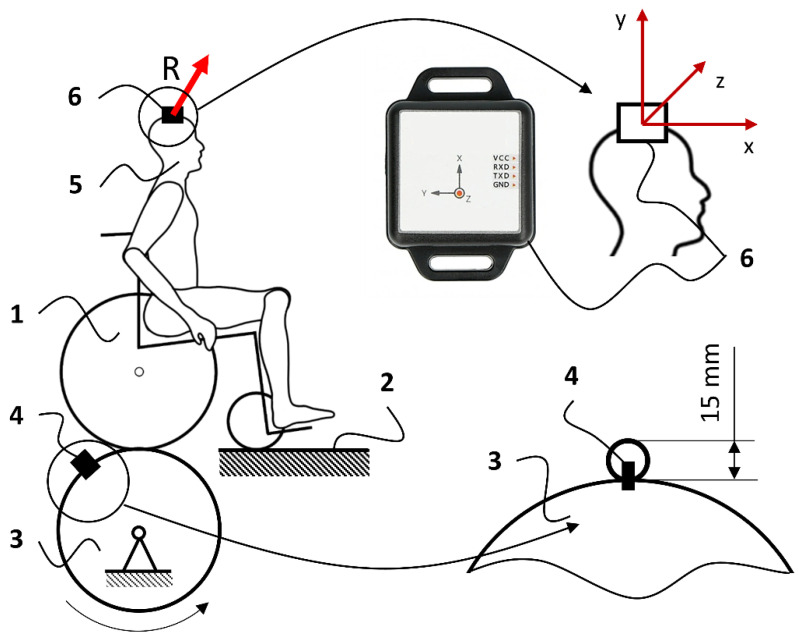
Diagram of the measurement system used, where 1—tested wheelchair, 2—stationary surface stabilizing the wheelchair, 3—cylinder supporting and propelling the rear wheels of the wheelchair, 4—transverse inequality generating kinematic excitation, 5—human exposed to vibrations, 6—accelerometer.

**Figure 3 materials-17-00641-f003:**
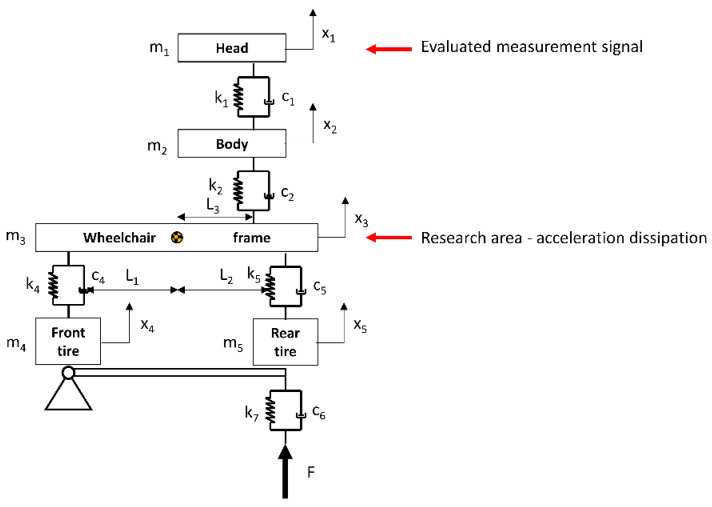
The schematic diagram of the analytical model, taking into account the segment for which the research was conducted, where m_i_—mass of the segment, k_i_—stiffness constant, c_i_—damping constant, L_3_—distance from the chair center to the human center, L_2_—distance from the chair center to the rear side, L_1_—distance from the chair center to the front, x_i_—displacement of the segments, F—forcing function.

**Figure 4 materials-17-00641-f004:**
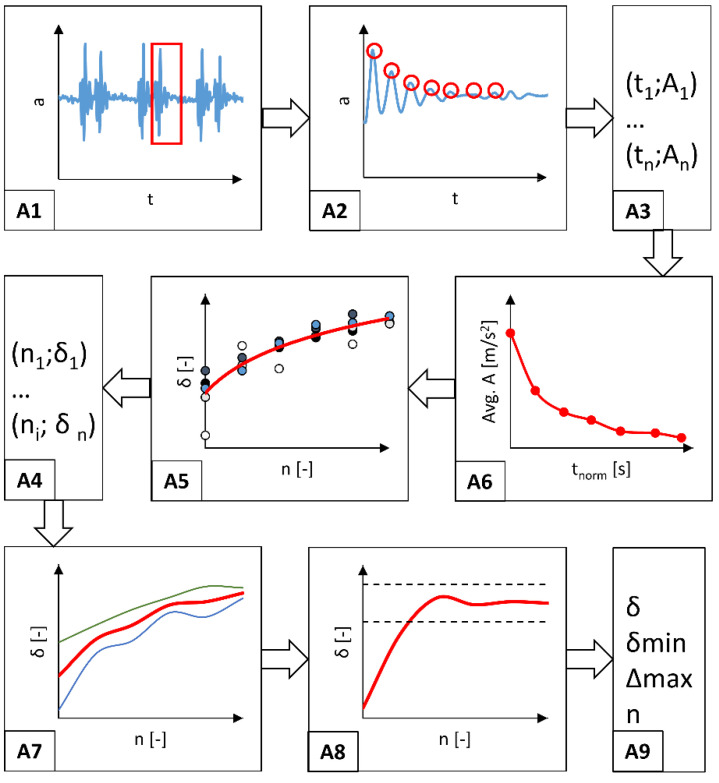
Schematic representation of the algorithm for processing the measurement signal recorded using the accelerometer, where: A1–A9 are steps of the algorithm (description of steps included in the text).

**Figure 5 materials-17-00641-f005:**
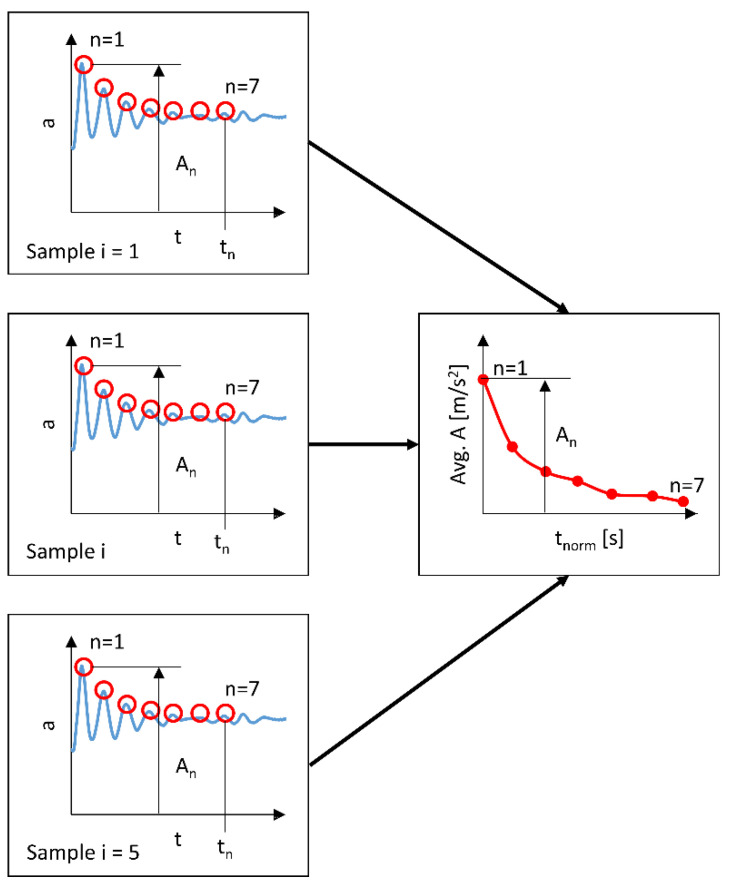
Diagram of extracting amplitudes of head acceleration *a* from the consistent cycles of kinematic excitation and method of measuring the average value of the average amplitude *A*.

**Figure 6 materials-17-00641-f006:**
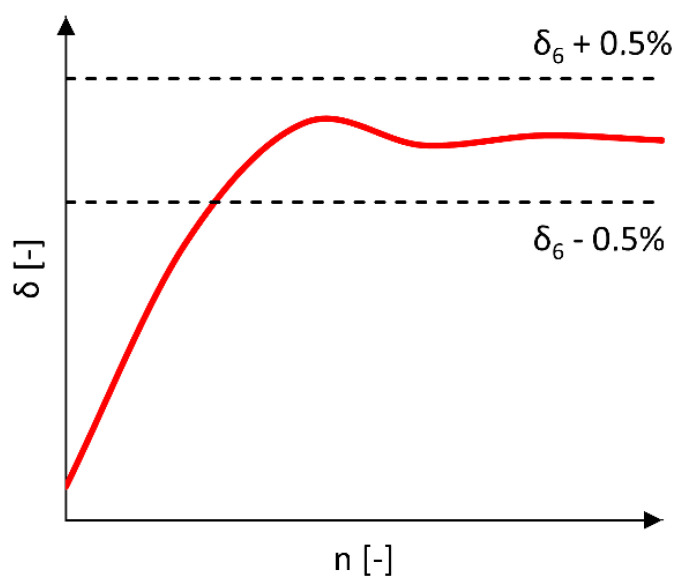
Diagram of the interval used to analyze the stabilization of the *δ* damping decrement.

**Figure 7 materials-17-00641-f007:**
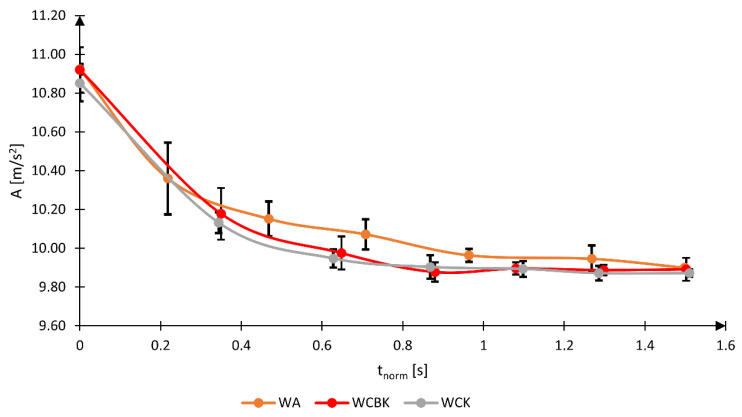
Graph of the change in amplitude as a function of normalized time with the limits of the confidence interval marked for the significance level *p* = 0.05 (WA—a wheelchair with an aluminum frame, WCBK—a wheelchair with a carbon fiber frame with a standard seat, WCK—a wheelchair with a frame made of carbon fiber and with a bucket seat, *t*_norm_—normalized value of the duration of the analyzed vibrations, A—vibration amplitude).

**Figure 8 materials-17-00641-f008:**
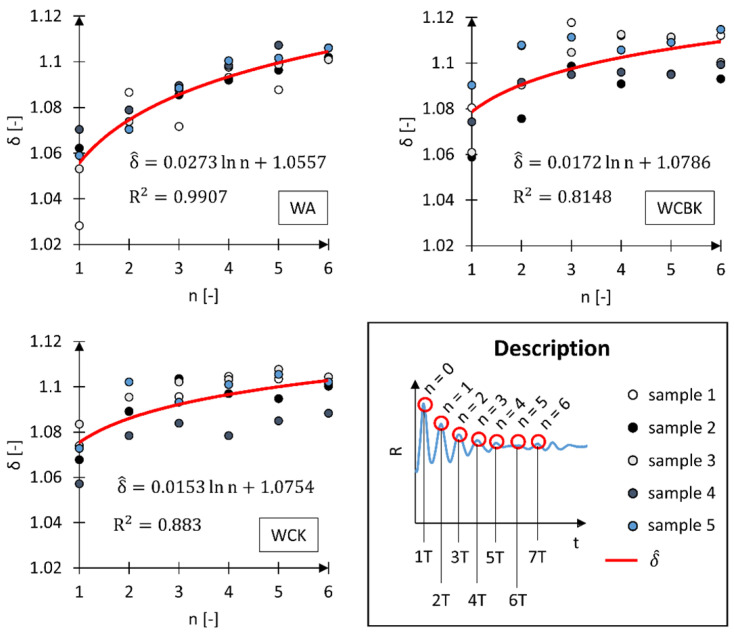
Dependence of the damping decrement *δ* relative to the first oscillation (*n* = 0) for a wheelchair with an aluminum alloy frame (WA), an active wheelchair with a carbon fiber frame with a standard seat (WCBK), and an active wheelchair with a carbon fiber frame with bucket seat (WCK). *δ*—damping decrement, $\hat{\delta }$—function characterizing the variability of the average value of the damping decrement, *n*—oscillation number, *T*—period, R^2^—coefficient of determination.

**Figure 9 materials-17-00641-f009:**
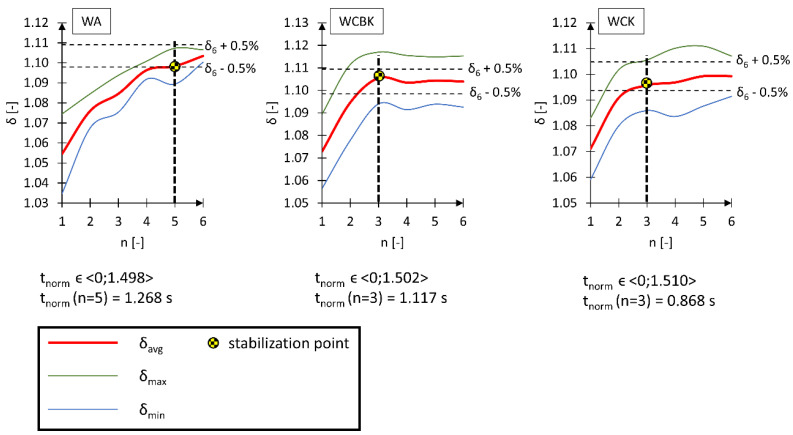
Graphs of the average value of the δ_avg_ damping decrement variability for an active wheelchair with an aluminum frame (WA), an active wheelchair with a carbon fiber frame with a standard seat (WCBK), and an active wheelchair with a carbon fiber frame with a bucket seat (WCK). δ_avg_—average damping decrement, *δ*_max_—maximum damping decrement, *δ*_min_—minimum damping decrement, *t*_norm_—normalized time of the observed damping phenomenon of the kinematic excitation, *n*—number of the observed oscillation of the kinematic deflection.

**Table 1 materials-17-00641-t001:** Morphological matrix of the tested wheelchair variants.

Variant Denotation	WA	WCK	WCBK
	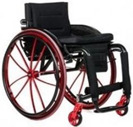	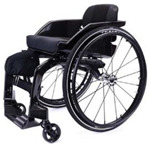	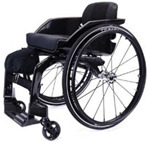
Driving wheel	pneumatic, 36 spokes, pressure 5 bar, diameter 24”	pneumatic, 36 spokes, pressure 5 bar, diameter 24”	pneumatic, 36 spokes, pressure 5 bar, diameter 24”
Caster wheel	full, diameter 4”	full, diameter 4”	full, diameter 4”
Seat cushion	pneumatic—Prevent Classic (AR-093)	pneumatic—Prevent Classic (AR-093)	pneumatic—Prevent Classic (AR-093)
Frame material	aluminum frame	carbon fiber frame	carbon fiber frame
Seat design	classic on stripes	bucket	classic on stripes
Overall mass	11.6 kg	8.2 kg	7.8 kg
Model; manufacturer	Aviator; MTB Poland, Łódź, Poland	Freeasy Bucket; Cosmotech LLC, Bytom, Poland	Freeasy Classic; Cosmotech LLC, Bytom, Poland

**Table 2 materials-17-00641-t002:** Amplitude values for a time interval equal to seven times the vibration period for three compared wheelchair variants: WA—a wheelchair with an aluminum frame, WCBK—a wheelchair with a carbon fiber frame with a standard seat, WCK—a wheelchair with a carbon fiber frame and bucket seat. Where: *T*—average vibration period with a confidence interval, *t*_norm_—normalized value of the duration of the analyzed vibrations, A—vibration amplitude at subsequent amplitudes.

		WA				WCBK				WCK		
*n*	*T*	*t* _norm_	*A*	±	*T*	*t* _norm_	*A*	±	*T*	*t* _norm_	*A*	±
	(s)	(s)	(m/s^2^)	(m/s^2^)	(s)	(s)	(m/s^2^)	(m/s^2^)	(s)	(s)	(m/s^2^)	(m/s^2^)
	0.250 ± 0.031	0	10.92	0.03	0.250 ± 0.063	0	10.92	0.12	0.252 ± 0.057	0	10.85	0.09
1		0.218	10.36	0.19		0.350	10.18	0.13		0.344	10.13	0.05
2		0.468	10.15	0.09		0.648	9.98	0.09		0.628	9.95	0.05
3		0.708	10.07	0.08		0.880	9.88	0.05		0.868	9.90	0.06
4		0.964	9.96	0.03		1.080	9.90	0.03		1.098	9.89	0.04
5		1.268	9.95	0.07		1.298	9.89	0.03		1.286	9.87	0.04
6		1.498	9.90	0.01		1.502	9.89	0.06		1.510	9.87	0.02

## Data Availability

Data are contained within the article.

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
