# Peer review of "Investigating the Impact of Carbon Fiber as a Wheelchair Frame Material on Its Ability to Dissipate Kinetic Energy and Reduce Vibrations"

_materials, 2024, doi:10.3390/ma17030641_

Round 1
Reviewer 1 Report
Comments and Suggestions for Authors
Overall, an interesting research topic.
However, some suggestions/questions are below:
1. Although the results suggest a significant improvement in the reduction of vibrations felt by the user, are they significant enough such that the reduction in vibration is able to be differentiated by the user between the two types of materials? It would've been nice to have the wheelchair user complete a questionnaire rating their comfort level and/or effect of vibration on themselves.
2. One concern is the wheelchair testing mechanism itself. The researchers placed the front casters of the tested wheelchairs onto a stationary platform. It would've been more realistic to use a testing mechanism such as a multi drum test machine such as that specified in the testing of wheelchairs in section 8 of the RESNA WC-1 standards testing. The overall resulting amplitudes using the methods outlined in this manuscript most likely would result in lower amplitudes being generated by the testing equipment due to the fact that only the main drive wheels were in motion rather than all four wheels of the wheelchair. If the caster wheels were nonstationary and instead being rotated at the same speed, the amplitudes measured by the accelerometer could possibly be higher and thus have a much larger significant difference between the two types of materials being tested.
3. Another questionable research method is the speed at which the tests were conducted. The authors state the simulated linear speed was approximately 2.28 km/h. Why was this speed chosen? This speed seems rather slow for the average manual wheelchair user as the average speed of a walking human is roughly 5 km/h. Another interesting set of tests would be to have conducted additional tests at different speeds to evaluate the capability of the two types of materials to absorb vibrations throughout a range of speeds.
Author Response
Thank you for your review.
Detailed answers in the attachment.

Reviewer 2 Report
Comments and Suggestions for Authors
The influence of the use of carbon fiber in the construction of wheelchair frames on the ability to dampen the energy of kinetic excitation research article deals with the comparative analysis of a wheelchair by using aluminum alloy frame with a frame made of carbon fiber due to the reduction of the scalar value and time of acceleration acting on the human body.
The article presents results of a vibration analysis. There is no scientific experimentation, so it should be treated more as a technical dissemination rather than a research article
Authors must make some modifications prior to publication.
1) L77 define MTB
2) The authors should specify more accurately who the manufacturers of the wheelchairs used in the study are.
3) Table 1. Add the total weight of each of the three wheelchairs
4) Why does the shape of the aluminum wheelchair seat vary compared to carbon fiber wheelchairs?
5) What would be the effect of the weight of the wheelchairs on the results obtained? The authors should provide an explanation for this.
Author Response

(The authors gave the same response as above.)

Reviewer 3 Report
Comments and Suggestions for Authors
1-The authors must develop a mathematical model for the wheelchair, at least 2D, showing the spring, damper, and masses elements, input, and output of the system.
2-Considering the damping impact of the back and sitting cushion as factors affecting the head.
3-FEA is recommended.
Author Response

(The authors gave the same response as above.)

Round 2
Reviewer 3 Report
Comments and Suggestions for Authors
Dear Authors
There are 3 comments; #1 is very important; it is about the Mathematical model that is not incorporated, unfortunately. At least a 2D mathematical model with a comparison with some results has to be there.
Author Response
Thank you for your valuable comments, the article has been supplemented with a chapter on the model, with a detailed answer presented in the attachment.
